# Processing Method of Gearbox with Non-Circular Gear Train and Its Application in Rice Potted Seedling Transplanting Mechanism

**Xin Xu, Maile Zhou \*, Xuegeng Chen and Jiajia Yang**

School of Agricultural Engineering, Jiangsu University, Zhenjiang 222013, China
\* Correspondence: zhoumaile@126.com; Tel.: +86-15051140832

**Abstract:** A non-circular gear train has the advantages of compact structure, easy to achieve dynamic balance, and is often used to drive an end-effector to achieve complex trajectory and attitude. The gearbox is the key component of non-circular gear train, which has complex structure, and requires high precision of each wheel shaft hole and its relative position. Conventional methods often use numerical control process tools to process the gearbox with non-circular gear train. The processing cost is too high, and there are some problems, such as shaft hole deflection and it can be difficult to ensure the phase position. This research has proposed a processing method of the gearbox with non-circular gear train and developed the corresponding combination tool. Taking the gearbox of a rice potted seedling transplanting mechanism as an example, a special cutting tool with five boring tools was designed which can process five holes at one clamping, and greatly improve the efficiency and accuracy of gearbox processing. The processed gearbox and other parts were assembled into a rice potted seedling transplanting mechanism and assembled into a rice potted seedling transplanting process to conduct a seedling transplanting performance experiment. The results show that the processing of the gearbox with non-circular gear train satisfies the requirement of processing accuracy, and the transplanting mechanism can accurately complete seedling picking, transporting, planting, and other series of actions. The processing method is suitable for the large-scale manufacturing of transplanting mechanism. In this research, aiming at the double arm transplanting mechanism of high-speed rice potted seedling transplanting, the gearbox was analyzed, and a processing method of the gearbox was proposed.

**Keywords:** non-circular gear train; gearbox; transplanting mechanism; processing method; combination tool

## 1. Introduction

At present, there are two methods of rice planting. One is direct sowing and the other is transplanting [1–3]. Direct sowing makes the field growth cycle longer and transplanting is beneficial to get strong seedlings. Transplanting methods are mainly artificial transplanting and mechanical transplanting [4]; mechanical transplanting mainly has blanket seedling transplanting and potted seedling transplanting two ways. Blanket seedlings need to tear the seedling roots when transplanting, resulting in seedling injury, affecting the subsequent growth of seedlings [5]. Potted seedling transplanting directly transplants the cultivated seedlings with nutritional pot bodies into the field. Potted seedling transplantation seedlings without slow seedling stage does not hurt the root, has early tillering, the yield can be increased by more than 10% and prolong the growing period of rice, and is conducive to expanding the planting area of fine grain [6,7]. Therefore, rice potted seedling transplanting technology has been widely studied. Wang et al. [8] designed a rice potted seedling wide narrow row transplanting mechanism that was based on the "Figure 8" shaped transplanting track, which combines a non-circular gear and a bevel gear. Wu et al. [9] completed the design of a three arm rotary rice potted seedling transplanting

mechanism that was based on the reverse design method of local motion trajectory, making it meet the transplanting requirements of rice potted seedling height perpendicularity. Cai et al. [10] designed a variable row spacing potted seedling transplanting mechanism by referring to the walking rice transplanter to ensure the uniform spacing of rice potted seedling when transplanting. Yu et al. [11,12] developed a rotating rice potted seedling transplanting mechanism based on elliptical incomplete non-circular gears, and applied the self-developed intermittent mechanism. Zuo et al. [13] established a mathematical model on the MATLAB platform, and designed a potted seedling transplanting mechanism with non-circular gear planetary gear train based on the non-uniform spline curve. The mechanism obtained the non-circular gear pitch curve through the determined 13 value points. Zhou et al. [14] proposed a method to get the non-circular gear pitch curve by fitting the type value points of a Bezier curve. Based on this method, a Bezier gear high-speed rice potted seedling transplanting mechanism was designed, which reproduced the trajectory and attitude of rice transplanting.

The gearbox is the key part of the transplanting mechanism; it will be the shaft, bearing, gear, and other parts together with appropriate assembly relations, so that they work in harmony with each other in order to transfer power [15]. The processing precision of the gearbox housing directly affects the assembling precision and motion precision of the gearbox [16]. At present, in order to improve the processing precision of box parts, various processing methods have been put forward. Wang et al. [17] designed a special hydraulic clamping fixture, the use of "one side two pin" positioning mode, so that the gearbox can achieve self-alignment and self-positioning, as well as a clamping complete gearbox processing of a variety of processes, greatly improving the precision of processing [18]. Zhang et al. [19] constructed the "feature-step" and "feature-face" peripheral Boolean matrix of the step-sorting problem for the low carbon and low cost in the processing process of box parts and solved the step-sorting problem through a genetic algorithm. Liu et al. [20] designed a two-station vertical combination process tool which fixed the gearbox on the movable workbench with clamps, the process tool assembled special compound tool, and made the process tool meet the needs of different aperture processing through the fine-tuning mechanism.

At present, most of the rice potted seedling transplanting mechanisms that have been developed at home and abroad use non-circular gear train mechanism, which is difficult to optimize the design. The gearbox, the core part of transplanting mechanism, is difficult to process, and it is difficult to achieve mass production. In this paper, aiming at the double arm transplanting mechanism of high-speed rice potted seedlings, the gearbox, the core component of the transplanting mechanism, is analyzed, and a processing method of a non-circular gear train gearbox is proposed.

## 2. Mechanism Analysis and Gearbox Introduction of Rice Potted Seedling Transplanting Mechanism

### 2.1. Composition of Rice Potted Seedling Transplanting Mechanism

The structure diagram of the double-arm transplanting mechanism of high-speed rice potted seedlings is shown in Figure 1. The transplanting mechanism of rice potted seedlings consists of a non-circular gear planetary gear train and two transplanting arms. The non-circular gear planetary gear train is composed of five non-circular gears and gearbox (planetary carrier). The tooth profile parameters of planetary gear I, planetary gear II, and solar gear are the same, and the intermediate gear I and intermediate gear II are the same conjugate non-circular gear. The solar gear is fixedly connected with the rack, the intermediate gear I and the intermediate gear II are engaged with the solar gear at the same time, the planetary gear I and the intermediate gear I are engaged with each other, the planetary gear II and the intermediate gear II are engaged with each other, the transplanting arm I and the transplanting arm II are fixedly connected with the planetary gear I and the planetary gear II, respectively. During working, the planetary carrier rotates clockwise with uniform speed and rotates unequally through the non-circular planetary gear train,

and the two transplanting arms move with uniform circular speed with the gearbox on the one hand, and rotate unequally with the planetary gear relative to the gearbox on the other hand. The combination of the two movements forms the trajectory and attitude that is required for rice transplanting.

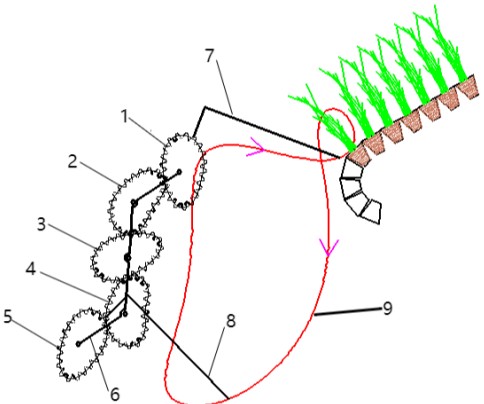

**Figure 1.** Schematic diagram of rice bowl seedling transplanting mechanism. 1, planetary gear I; 2, intermediate gear I; 3, solar gear; 4, intermediate gear II; 5, planetary gear II;6, planetary carrier; 7, transplanting arm I; 8, transplanting arm II; 9, transplant trajectory.

The transplanting arm is mainly composed of the shell of the transplanting arm, clip seedling device, seedling piece, seedling pushing device, seedling pushing rod, CAM, and fork, as shown in Figure 2a. The CAM and planetary frame are fixed; the transplanting arm shell and planetary axis are fixed; the transplanting arm shell relative to the CAM rotation, fork, and transplanting arm shell are hinged on one end; and CAM action on one end of the transplanting rod relative to the transplanting arm shell do reciprocating linear motion. The seedling clamping device and the seedling pushing device are fixed on the seedling pushing rod. The seedling clamping device is arranged on the inside of a pair of seedling clamping pieces, and the seedling pushing device is arranged on the bottom of the seedling clamping piece. The working process is shown in Figure 2b. At the initial state, the seedling clamp supports two seedling pieces from the inside. Under the condition of taking seedlings, the seedling holder moves backward, and the two seedling pieces are closed under their own elasticity to tighten the middle seedlings. In the condition of seedling pushing, the seedling holder moves forward, pushes the two seedling pieces, and releases the seedlings in them. At the same time, the seedling holder acts on the seedling pot matrix and plants the seedlings into the soil.

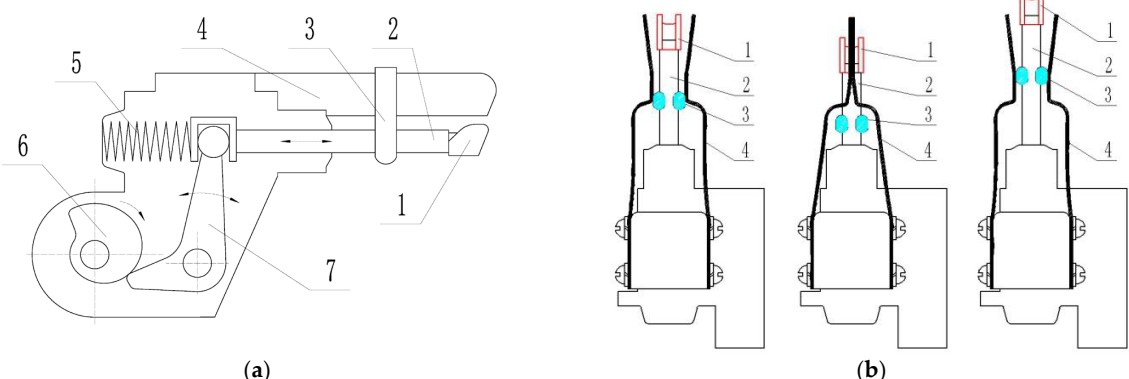

|(**a**)|(**b**)|

**Figure 2.** (**a**) Transplanting arm; (**b**) Clip seedling device. The following parts are shown: 1, seedling pushing device; 2, seedling pushing rod; 3, clip seedling device; 4, seedling piece; 5, push the seedling spring; 6, CAM; 7, fork.

### 2.2. The Trajectory and Attitude Analysis of the Transplanting Mechanism

In this paper, the planetary gear train is used in the transplanting mechanism of rice potted seedlings to realize three actions: seedling picking, transporting, and planting. Picking seedlings requires the transplanting mechanism to clamp the potted seedlings from the hole plate. Transportation requires that the transplanting mechanism keep the state of the seedling and transport the potted seedlings from the position of picking the seedlings to the position of the pushing seedlings. Planting requires the transplanting mechanism to plant the seedlings into the rice field.

The movement trajectory of the rice potted seedling transplanting mechanism is a "Figure 8" trajectory, as shown in Figure 3. In the "Figure 8" trajectory, the upper ring buckle completes seedling picking and the lower ring buckle completes transporting and planting. In the initial condition, the non-circular gear and the transplanting arm are installed according to the initial installation angle. The transplanting mechanism moves clockwise to the position of picking the seedlings, and the tip of the transplanting arm moves to the point of packing the seedlings (point A). Then, the potted seedlings are clamped by the transplanting arm and pulled out along the growth direction of the pot seedlings to realize the separation of pot seedlings and hole plate. From point A to point B, the transplanting arm is kept clamped, the potted seedlings are transported to the planting position, and the pot seedlings are inserted into the field in an upright state by turning at an angle (about the seedling box angle). After that, the transplanting arm completes the reset action (point B to point C), and is ready to complete the next seedling picking.

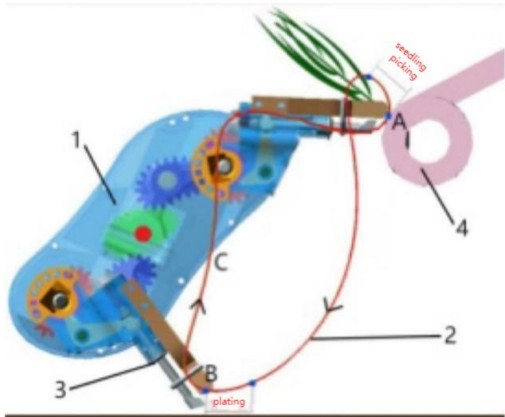

**Figure 3.** Seedling trajectory diagram. 1, gearbox; 2, transplanting trajectory; 3, transplanting arm; 4, hole plate.

The non-circular gear planetary gear train is the core working component to realize the rice potted seedling transplanting mechanism. Through the non-circular gear planetary gear train's unequal speed transmission characteristics, the transplanting arm can obtain the complex trajectory and attitude that is required for rice potted seedling transplanting. In the double arm transplanting mechanism of high-speed rice potted seedlings, the gearbox is the original moving part, and the rotation of the gearbox drives the movement of the non-circular gear planetary gear train to ensure that the transplanting arm can complete the movement of rice seedling picking, transporting, and planting. In order to ensure the accuracy of the trajectory and that is attitude required by the rice potted seedling transplanting mechanism for rice transplanting, the design and processing of the gearbox is particularly important.

### 2.3. Gearbox Design

The gearbox is a key part of the non-circular gear train, which contains five holes with relative position precision requirements (see Figure 4), namely, planetary shaft holes I and II, intermediate shaft holes I and II, and solar shaft holes. The five holes form a parallel hole system with centro-symmetry. The intermediate shaft hole I, intermediate shaft hole II,

with the solar shaft hole form a straight line. There is a corner between the planetary shaft hole I and the intermediate shaft hole I and the intermediate shaft hole I and the solar shaft hole at an angle of 127°. Planetary shaft hole I and planetary shaft hole II have the same size, and intermediate shaft hole I and intermediate shaft hole II have the same size. The center distance of each shaft hole is the same, which is 51.42mm. The specific dimensions of the shaft hole are shown in Table 1.

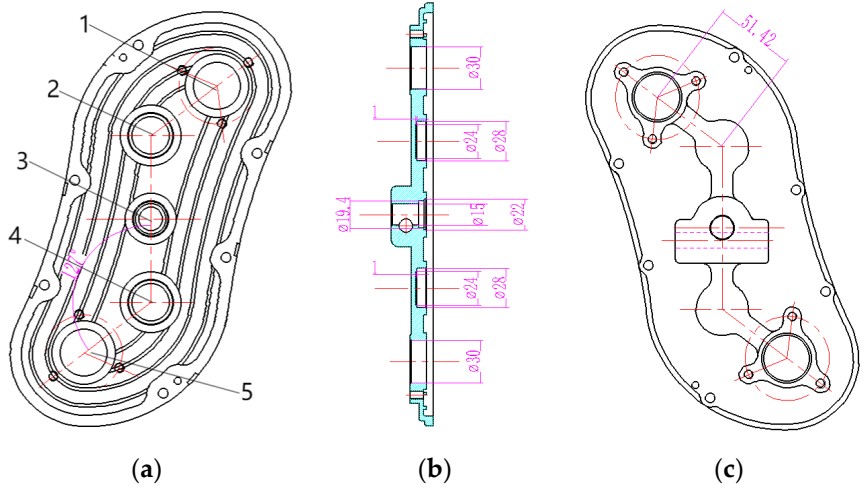

|         (a)          |          (b)          |          (c)          |

**Figure 4.** (**a**) Gearbox front view; (**b**) gearbox sectional view; (**c**) gearbox back view. The following parts are shown: 1, planetary shaft holes I; 2, intermediate shaft holes I; 3, solar shaft hole; 4, intermediate shaft holes II; 5, planetary shaft holes II.

**Table 1.** Gearbox shaft hole size.

| Hole Name | Diameter (mm) | Depth (mm) |
| --- | --- | --- |
| Planetary shaft holes | 30 | 12 |
| Intermediate shaft holes | 28 | 7 |
|  | 24 | 1 |
|  | 22 | 2 |
| Solar shaft hole | 19.4 | 3.6 |
|  | 15 | 16.4 |

## 3. Gearbox Processing Tool and Processing Method

### 3.1. Gearbox Processing Tool Design

It is necessary to ensure the processing precision of the gearbox when mass manufacturing the double arm transplanting mechanism for high-speed rice potted seedlings. The shape of the gearbox is complex, and the requirements for the position precision of its five shaft holes are high. The processing quality of the gearbox not only affects the assembly precision and motion precision, but also affects the precision of the transplanting arm to achieve the predetermined trajectory and attitude [21]. When the processing error of the gearbox's five shaft holes is too large, the gear cannot engage correctly or the backlash is too large when assembling the non-circular gear, which leads to the failure of the rice potted seedling transplanting mechanism to carry out transplanting actions such as seedling picking, transporting, and planting.

In this paper, the errors that may occur in the processing process are simulated in VB6.0, and the parameters such as the center distance and angle of the axle hole of the gearbox are changed, and the trajectory and attitude of the rice seedling transplanting mechanism are visually displayed at this time. As shown in Figures 5 and 6, when the distance between the center of the corner and shaft hole is too large due to the machining error of the gearbox, the seedling picking track will be greatly deviated, and the interference between the transplant arm and the hole plate may occur, and the seedling picking point

is too high or too low, so that the seedling picking action cannot be completed at the seedling picking point. It will greatly affect the success rate of taking seedlings and it may even occur that the transplant mechanism of rice pot seedlings cannot complete the transplanting actions such as taking seedlings, conveying, and planting. In view of the above phenomenon, in order to reduce or even eliminate the reduction of the performance of taking the seedling that is caused by the machining error of the gearbox, this paper designed a special tool for machining the gearbox.

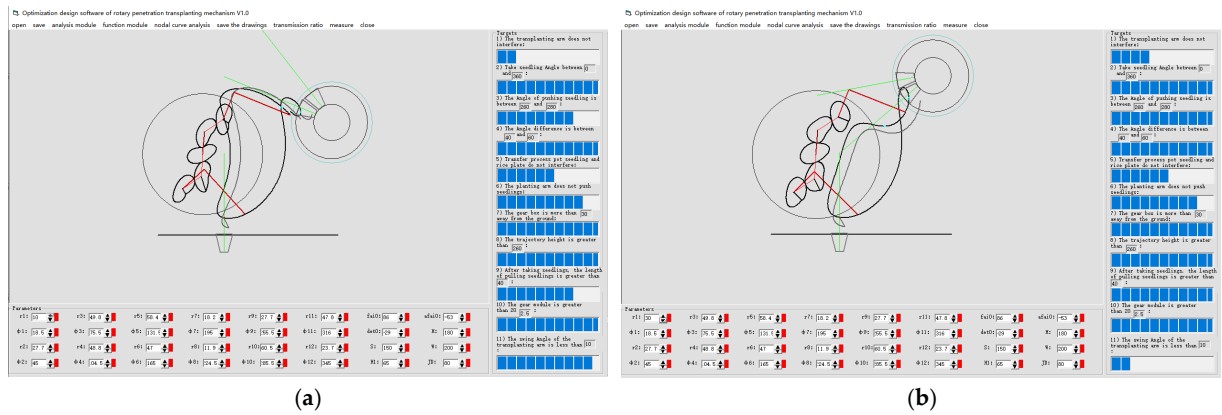

**Figure 5.** (**a**) transplanting trajectory with decreasing center distance; (**b**) transplanting trajectory with increasing center distance.

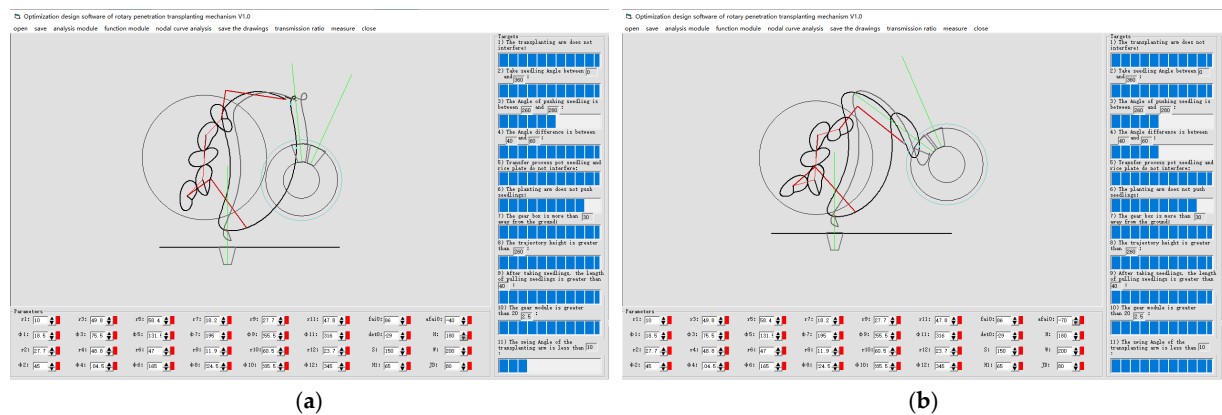

**Figure 6.** (**a**) transplanting trajectory with decreasing corner; (**b**) transplanting trajectory with increasing corner.

As the gearbox contains five shaft holes that need to be processed, the ordinary boring process needs to be clamped repeatedly in the process of boring. Many clamps are not only laborious, but also need to be repositioned after each processing of a shaft hole, which greatly improves the error and reject rate. For initial casting of a gearbox blank of possible problems such as pore size accuracy and surface roughness and the gearbox of batch processing, we designed a five hole processing tool as shown in Figure 7. Clamping can realize five shaft hole processing, reduce processing datum transformation, improve the processing precision, and efficiency of gearbox. The processing tool mainly includes five power shafts, limit shafts, power commutation shafts, etc. The five power shafts are installed and positioned according to the relative position of each shaft hole in the gearbox, as shown in Figure 8. The solar shaft, intermediate shaft I and intermediate shaft II, and planetary shaft I and planetary shaft II rely on the lock nut, side plate, and limit shaft to achieve limit. The boring bar and five shafts are connected by thread, and a power reversing shaft is equipped with a spur gear. When working, the power is introduced from the solar shaft, and transmitted to the intermediate shaft and the planetary shaft through the spur gear on the reversing shaft, driving the corresponding boring tool to

rotate uniformly. According to the size of the boring hole, the processing tool can loosen the positioning bolts on the boring bar, so as to freely adjust the extension of the boring tool to realize the processing of different hole diameters. The gearbox solar shaft hole is the deepest in the processing of the need for priority processing, so the length of the solar shaft boring tool is slightly longer than the other four boring tools.

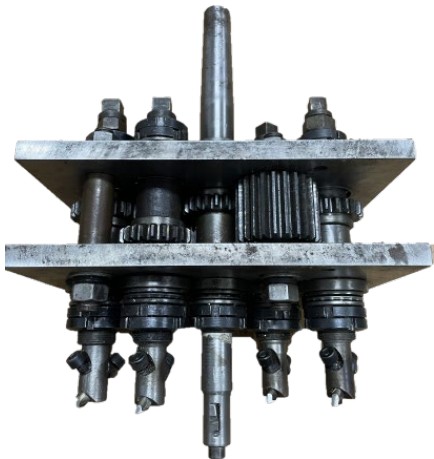

**Figure 7.** Processing tool drawings.

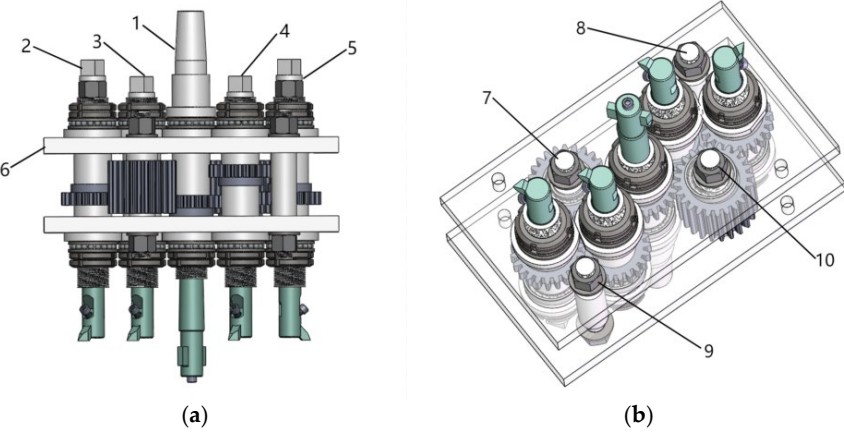

(**a**)                                    (**b**)

**Figure 8.** (**a**) Front view of the processing tool; (**b**) side view of processing tool shaft. The following parts are shown: 1, solar shaft; 2, planetary shaft I; 3, intermediate shaft I; 4, intermediate shaft II; 5, planetary shaft II; 6, support side plate; 7, power reversing shaft I; 8, limit shaft I; 9, limit shaft II; 10, power reversing shaft.

### 3.2. Processing Method Design

The gearbox processing and boring process is shown in Figure 9, mainly including processing tools, tool support, positioning base, and special fixtures. When processing the gearbox, according to the size of the processing tool, a tool support is installed on the boring process, so that the processing tool is fixed stably above the gearbox, and the solar shaft of the processing tool is fixedly connected with the spindle of the boring process. At the same time, in order to facilitate the clamping of the gearbox, a positioning base was designed. The positioning base refers to the boss on the outer surface of the gearbox. According to the maximum diameter and relative position of the boss, five holes of corresponding size are processed on the positioning base. During clamping, only the casting rough of the gearbox is placed on the positioning base one by one according to the hole position, and then the gearbox and the positioning base are fixed on the workbench of the boring process with a special fixture. This ensures the relative position between the boring tool and the gearbox during processing, and greatly reduces the difficulty in positioning and clamping the gearbox.

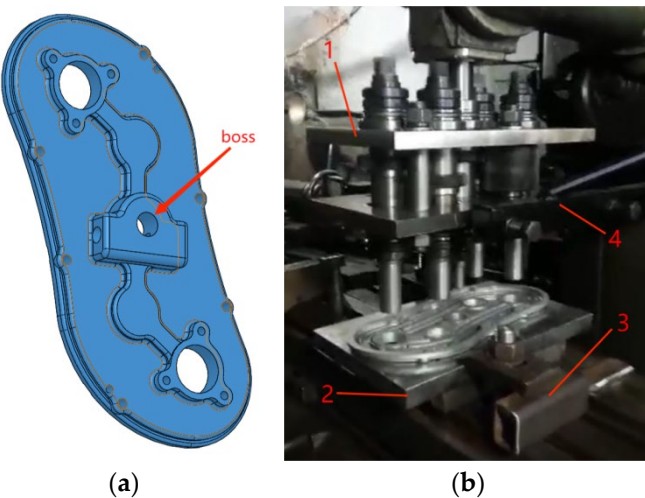

(**a**)  (**b**)

**Figure 9.** (**a**) Three dimensional diagram of the gearbox; (**b**) boring process processing. The following parts are shown: 1, processing tools; 2, positioning base; 3, special fixture, 4, tool support.

When processing the gearbox, the gearbox is fixed on the workbench, the boring tool keeps rotating under the action of the boring process spindle, and the combined tool moves downward for feeding. Since the solar shaft hole is deep, the solar shaft hole is processed first, then the intermediate shaft hole and the planetary shaft hole are processed; the processing diagram is shown in Figure 10. The aperture size of the processing method is not limited by the tool size, and has strong error correction ability, which can effectively correct the deflection of the original hole shaft and reduce the surface roughness of the shaft hole.

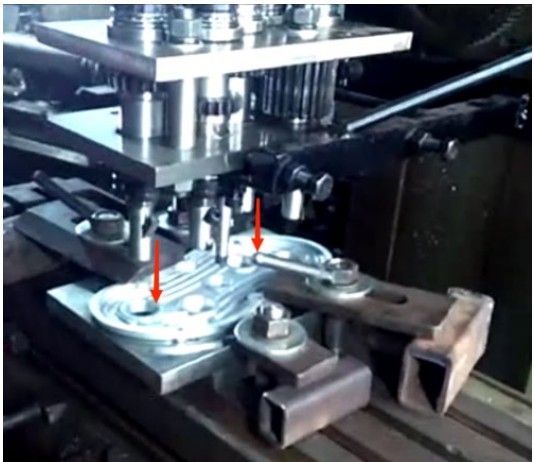

**Figure 10.** Gearbox processing diagram.

## 4. Verification Experiment

In order to verify the correctness of the five-hole processing method of the gearbox that was proposed, the processed gearbox was installed on the rice potted seedling transplanting process for seedling transplanting performance experiment, as shown in Figure 11a. The plastic hole plate was used in the experiment, with 14 holes in the transverse direction and 29 holes in the longitudinal direction, and the distance between the holes was 20mm. On average, there are one to three seedlings per hole. The rice seedlings grow well without root connection. The seedling age is 30 days. In the experiment of picking seedlings, the efficiency of taking seedlings was 150 holes/min, the transplanting mechanism operated smoothly, and the success rate of taking seedlings was 96.4%. In order to further verify the correctness of the gearbox processing method, the field transplanting performance experiment of the transplanting mechanism was carried out, as shown in Figure 11b. The

test showed that the processed gearbox can meet the design requirements of rice potted seedling transplanting mechanism, and the transplanting arm can accurately realize the actions of seedling picking, transporting, and planting. Therefore, the proposed five-hole processing method can efficiently and conveniently process the gearbox to meet the requirements, and can be used to realize the mass processing of a gearbox.

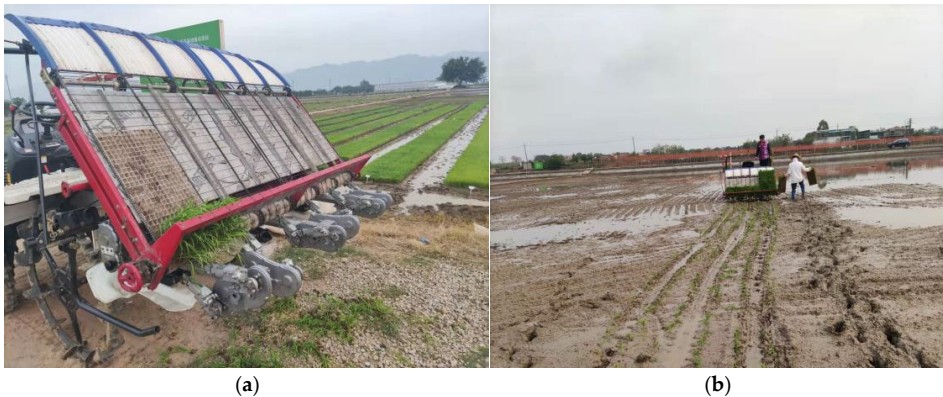

(**a**)                                                     (**b**)

**Figure 11.** (**a**) Seedling picking experiment; (**b**) field experiment.

## 5. Conclusions and Prospect

The gearbox of rice potted seedling transplanting mechanism is complex in structure, and the precision of each shaft hole and its relative position is high. Conventional processing methods are difficult to achieve for batch processing, and the processing cost is too high, easy to exist in the shaft hole deflection, difficult to ensure the relative position, and other problems. In this research, a processing method of a non-circular gearbox was proposed, and a special tool with five boring tools was developed which can process five holes at one clamping, greatly improving the processing efficiency and precision of the gearbox. A seedling transplanting experiment were carried out, the results showed that the transplanting mechanism worked smoothly and the success rate of seedling picking was 96.4% when the seedling collection efficiency was 150 holes/min.

At present, the processing method on the rice pot seedling transplanting mechanism processing has been a complete success, but in the interest of time, did not use the tool for the machining of other similar parts. In the subsequent period, an extension of the need to promote the processing cutting tool using the machining method, complete vegetable transplanting mechanism, and other crop transplanting mechanisms of batch processing

**Author Contributions:** Conceptualization, X.X. and M.Z.; methodology, X.X.; writing—original draft preparation, X.X.; writing—review and editing, M.Z., X.C. and J.Y. All authors have read and agreed to the published version of the manuscript.

**Funding:** This study was financially supported by the National Natural Science Foundation of China (Grant No. 52005221), Natural Science Foundation of Jiangsu Province (Grant No. BK20200897), Jiangsu Agriculture Science and Technology Innovation Fund (Grant No. CX(22)3089), China Post-doctoral Science Foundation (Grant No. 2021M691315), Key Laboratory of Modern Agricultural Equipment and Technology (Jiangsu University), High-Tech Key Laboratory of Agricultural Equipment and Intelligence of Jiangsu Province, and Priority Academic Program Development of Jiangsu Higher Education Institutions (Grant No. PAPD-2018-87).

**Institutional Review Board Statement:** Not applicable.

**Informed Consent Statement:** Not applicable.

**Data Availability Statement:** All of the data that were generated or analyzed during this study are included in this published article.

**Conflicts of Interest:** The authors declare no conflict of interest.

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
