# Peer review of "Processing Method of Gearbox with Non-Circular Gear Train and Its Application in Rice Potted Seedling Transplanting Mechanism"

_agriculture, doi:10.3390/agriculture12101676_

Round 1

Reviewer 1 Report (New Reviewer)

The research content has certain practical value, and it is suggested to revise as follows:

1. The same terminology should be the same, such as “hole plate” and “soft disc”.

2. The non-circular gear planetary gear train of the transplanting mechanism is explained in detail, but the structure of the transplanting arm is not described too much, so it is suggested to add.

3. In the third chapter, the specific changes of center distance and corner should be marked, so as to highlight the change of seedling trajectory when the processing error becomes large.

4. Some language expressions in the text should be modified appropriately

Author Response

Reviewer 2 Report (New Reviewer)

The proposed processing method and the designed processing tool meet the processing requirements of low cost and large quantities in the field of agricultural machinery, and also ensure the processing accuracy, which has a great contribution to the field of pot seedling transplanting. The article should make some modifications, and the suggestions are as follows:

1. Design and processing are finished work, grammar should use the past tense.

2. In chapter 3, the manuscript only mentions the deviation of the seedling selection trajectory caused by the error, without analyzing the seedling selection trajectory formed by the simulation error in VB 6.0. It is necessary to explain why these trajectories cannot complete the seedling selection.

3. The intermediate shaft hole and the solar shaft hole of the gearbox are holes with multiple diameters. How do the machining tools realize the machining of such holes?

4. The labeling in Figure 3 is not rigorous enough, so it is suggested to make minor modifications.

5. The format of Figure 5 is wrong and the 6 pictures should have the same format.

Author Response

Reviewer 3 Report (New Reviewer)

This paper proposed processing a unique gearbox using a special cutting tool with five boring tools to process five holes at one clamping and improve the efficiency and accuracy of gearbox processing and evaluated this gearbox in the rice seedling transplanting mechanism.

Some comments to the authors:

1. In line 85, the authors should revise the numbering of the headings throughout the manuscript.

2. In Figure 3, the dimensions under this figure did not mention; please add the dimensions in mm.

3. In section 3, the authors should mention the name and the version of the software used in this section.

4. In Figure 8, please amend the brightness of this figure, and the numbers are not clear in this figure.

5. In the conclusion section, the authors should mention their future perspectives for this study and its various future agricultural applications.

6. Please revise the reference format in the references list according to the journal’s style.

Author Response

Reviewer 4 Report (New Reviewer)

The processing method proposed in this article is of great significance to batch processing of transplanting mechanism with non-circular gear train, which can improve machining accuracy and efficiency and reduce machining cost. However, there are some inconsistencies in the professional language used in this article, and unified language is needed. Two line segments are annotated in Figure 2, but are not explained. In Figure 5, the main parameters that cause the trajectory to change are to be externalized.

Author Response

This manuscript is a resubmission of an earlier submission. The following is a list of the peer review reports and author responses from that submission.

Round 1

Reviewer 1 Report

The article is not of a scientific nature. It applies to a simple engineering solution, without basic research. However, the article is important for the purposes of its practical application.

Author Response

Point 1: Does the introduction provide sufficient background and include all relevant references?

Response : For the question raised by the reviewer, the latest references on rice pot seedling transplanting institutions are added in the introduction. The corresponding modifications have been shown in red font.

Point 2: Is the research design appropriate?

Response: At present, most of them use CNC machine tools to process box type parts. The cost of CNC processing is high, and it is not suitable for mass production, which does not meet the requirements of low-cost processing in the field of agricultural machinery. This paper proposes a special processing method and designs a combined cutter for the processing of the gearbox of rice transplanting mechanism (not limited to rice transplanting), which can meet the processing requirements of agricultural machinery at low cost and ensure the processing accuracy in mass production.

Point 3: Are the methods adequately described?

Response: For the question raised by the reviewer, 2.1.2 (Trajectory and posture analysis of transplanting mechanism) is added in Chapter 2 to analyze in detail the "8-shaped" trajectories of rice potted seedling transplanting mechanism and the posture of transplanting arm under the conditions of picking, transporting and planting seedlings.

During gearbox processing, it is easy to have errors in corners and shaft hole center distance. If the error is too large, it will lead to large deviation in seedling picking track, reduce the success rate of seedling picking, and even fail to achieve seedling picking, transportation, planting and other actions. To solve this problem, in VB6.0, this paper simulates the errors that may occur in the processing process, changes the parameters such as the center distance of the gearbox shaft hole and the corner, and visually shows the track and posture of the rice seedling transplanting mechanism at this time. The specific description is added in Chapter 3.

All new contents are marked in red.

Reviewer 2 Report

The paper shows an application of the non-circular planetary gear trains in rice bowl seedling. 

The paper is more a descriptive presentation without scientific description of the realised function and trajectory. The literature research is limited geographically. The only one contribution seems to be the processing tool.

Author Response

Point 1: Does the introduction provide sufficient background and include all relevant references? point 2: Are all the cited references relevant to the research?

Response : For the questions raised by the reviewer, the latest references on rice pot seedling transplanting institutions are added in the introduction. The corresponding modifications have been shown in red font.

Point 3: Is the research design appropriate?

Response: At present, most of them use CNC machine tools to process box type parts. The cost of CNC processing is high, and it is not suitable for mass production, which does not meet the requirements of low-cost processing in the field of agricultural machinery. This paper proposes a special processing method and designs a combined cutter for the processing of the gearbox of rice transplanting mechanism (not limited to rice transplanting), which can meet the processing requirements of agricultural machinery at low cost and ensure the processing accuracy in mass production.

Point 3: Are the methods adequately described?

Response: For the question raised by the reviewer, 2.1.2 (Trajectory and posture analysis of transplanting mechanism) is added in Chapter 2 to analyze in detail the "8-shaped" trajectories of rice potted seedling transplanting mechanism and the posture of transplanting arm under the conditions of picking, transporting and planting seedlings.

During gearbox processing, it is easy to have errors in corners and shaft hole center distance. If the error is too large, it will lead to large deviation in seedling picking track, reduce the success rate of seedling picking, and even fail to achieve seedling picking, transportation, planting and other actions. To solve this problem, in VB6.0, this paper simulates the errors that may occur in the processing process, changes the parameters such as the center distance of the gearbox shaft hole and the corner, and visually shows the track and posture of the rice seedling transplanting mechanism at this time. The specific description is added in Chapter 3.

For question 3, all the new content is marked in red font.

Point 4: Are the results clearly presented?

Point 5: Are the conclusions supported by the results?

Response: For the questions raised by the reviewer, he author assembled the gear box processed by this processing method on the rice potted seedling transplanting mechanism, and repeatedly carried out several seedling taking tests and field tests. When working, the transplanting mechanism runs smoothly, and the success rate of picking seedlings reaches 96.4%, which fully verifies the correctness of the gear box processing method. The specific seedling performance test is presented in Chapter 4.
